# Comparative Analysis of Injection of Pyrolysis Oil from Plastics and Gasoline into the Engine Cylinder and Atomization by a Direct High-Pressure Injector

Magdalena Szwaja [1], Jeffrey D. Naber [2], David Shonnard [3], Daniel Kulas [3], Ali Zolghadr [3] and Stanislaw Szwaja [1,*]

[1] Department of Thermal Machinery, Faculty of Mechanical Engineering and Computer Science, Czestochowa University of Technology, Dabrowskiego 69, 42-200 Czestochowa, Poland

[2] Department of Mechanical Engineering-Engineering Mechanics, Michigan Technological University, 1400 Townsend Drive, Houghton, MI 49931, USA

[3] Department of Chemical Engineering, Michigan Technological University, 1400 Townsend Drive, Houghton, MI 49931, USA

* Correspondence: stanislaw.szwaja@pcz.pl; Tel.: +48-885-840-483

**Abstract:** The article discusses the results of experimental studies on the course of pyrolysis oil injection through the high-pressure injector of a direct-injection engine. The pyrolysis oil used for the tests was derived from waste plastics (mainly high-density polyethylene—HDPE). This oil was then distilled. The article also describes the production technology of this pyrolysis oil on a laboratory scale. It presents the results of the chemical composition of the raw pyrolysis oil and the oil after the distillation process using GC-MS analysis. Fuel injection tests were carried out for the distilled pyrolysis oil and a 91 RON gasoline in order to perform a comparative analysis with the tested pyrolysis oil. In this case, the research was focused on the injected spray cloud analysis. The essential tested parameter was the Sauter Mean Diameter (SMD) of fuel droplets measured at the injection pressure of 400 bar. The analysis showed that the oil after distillation contained a significant proportion of light hydrocarbons similar to gasoline, and that the SMDs for distilled pyrolysis oil and gasoline were similar in the 7–9 µm range. In conclusion, it can be considered that distilled pyrolysis oil from HDPE can be used both as an additive for blending with gasoline in a spark-ignition engine or as a single fuel for a gasoline compression-ignition direct injection engine.

**Keywords:** distilled pyrolysis oil; plastics; injection; Sauter mean diameter; fuel

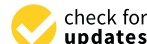



## 1. Introduction

In the search for alternative fuels, intensive studies have been focused on various liquid and gaseous by-products obtained from the thermal treatment of organic substances. Among others, the pyrolysis process is considered a promising technology for utilizing waste tires; however, hard-to-recycle plastics, such as polyethylene and polypropylene, can also be processed with this technology [1]. A major product of this processing is pyrolysis oil. Typical raw pyrolysis oil from a large-scale production line is a black substance with a strong characteristic odor, especially for waste tire feedstock. A typical pyrolysis oil consists of the following: light and heavy hydrocarbons and tars, water, and organic acids. As reviewed in the literature, plastics and waste tires are managed as valuable feedstock to a pyrolysis reactor for producing high-calorific liquids, known as pyrolysis oils [2,3]. Raw pyrolysis oils have been directly used as fuel for boilers for domestic heating systems. Due to its relatively high calorific value, pyrolysis oil can also be considered an alternative fuel for the internal combustion engine [4–6]. It is difficult to evaluate to what extent pyrolysis oil can be used either for a spark-ignition (SI) or a compression-ignition (CI) engine because pyrolysis oil contains both light and heavy hydrocarbons characterized

by various cetane and octane numbers. Among others, Hurdogan et al. observed that, due to blending pyrolysis oil of 10% with diesel fuel, the tested engine operated at its nominal torque and power output [4]. Furthermore, Tudu et al. also examined pyrolysis oil and tend to use this oil after blending with diesel fuel for the CI engine [5]. Umeki et al. found that the mixture of pyrolysis oil with diesel fuel has properties that allow them to be used as fuel for a CI engine [6]. On the other hand, the pyrolysis oil they tested had an octane number close to that of premium gasoline; however, it should be emphasized that this information is somewhat surprising if reviewed with other works in this field. In general, research in this field has been conducted on the combustion of pyrolysis oil in small amounts of 5–10% [4,7–9] and amounts higher than 20% [10,11] added to diesel fuel. Martinez et al. conducted an investigation on a blend consisting of 5% pyrolysis tire oil for the diesel-fueled turbocharged engine [9]. They observed higher specific fuel consumption and lower thermal efficiency, particularly at low loads. Whereas, at full loads, the engine did not show any deterioration in its performance. Koc et al. carried out a study on a diesel-fueled engine running on biodiesel-diesel blends with 5% and 10% tire pyrolysis oil [8]. They observed that the addition of 10% of tire pyrolysis oil in those blends contributed to a reduction in NOx and CO emissions. Regarding the higher pyrolysis oil content in diesel-fueled engines, Murugan et al. conducted tests on applying 70% pyrolysis oil in diesel fuel [11]. He found that the thermal efficiency decreased, but smoke, HC, and CO emissions increased. Further investigation in this area was carried out by Karagoz et al. [12]. They conducted engine tests with pyrolysis oil present in diesel blends at levels of 10, 30, and 50%. The results showed that the best engine load parameters were obtained when mixed with 10% pyrolysis oil. One can also find scientific papers on the combustion of only pyrolysis oil in the internal combustion engine [13,14]. Regarding fuels for stationary engines, Vihar et al. found that tire pyrolysis oil can be successfully applied to a heavy-duty turbocharged diesel engine [13]. Research work by Hurdogan et al. [4] and Karagoz et al. [10] lead to the conclusion that pyrolysis oil can be implemented in diesel engines; however, they supported their observations on the physicochemical properties of a typical pyrolysis oil from tire thermal processing.

According to a review of the literature, most of the research concerns the combustion of pyrolysis oil in a diesel engine. Regarding the state-of-the-art pyrolysis oil combustion in a SI engine, the literature database is relatively poor in comparison to diesel-fueled engines. An interesting study was performed by Sunaryo [15]. He conducted experimental studies focused on operation and exhaust emissions from a diesel engine powered by pyrolysis oil obtained from plastics thermal processing. Although his study was conducted in a diesel engine, he found that the tested oil was characterized by an octane rating of 88 to 92. Consequently, that pyrolysis oil was proposed as an alternative fuel for SI engines. Tests of pyrolysis oil in a spark-ignition gasoline engine were conducted by Kareddula and Puli [16]. They concentrated on engine performance and toxic exhaust emissions from the engine-fueled blends with 5%, 10%, 15% and 20% plastic pyrolysis oil. They observed an increase in NOx, but HC emissions were reduced. They came to the conclusion that the engine could run on mixtures consisting of maximal 20% pyrolysis oil. Szwaja et al. tested pyrolysis oil at an amount of 25% mixed with ethanol in the SI engine [17]. Even though, their engine worked at a compression ratio of 9.5, they found this blend did not cause knock despite the relatively low octane rating of the pyrolysis oil used for tests. Kareddula and Puli conducted tests on gasoline blends with pyrolysis oil of 15% and ethanol of 5% [16]. They analyzed performance and toxic exhaust emissions from their SI engine fueled with those blends. Unfortunately, they found that the engine's overall efficiency decreased and NOx emissions increased when compared to the tests with gasoline only. Hence, they recommended adding 5% ethanol for reduction of NOx emissions. Based on these selected publications, it can therefore be concluded that the research on pyrolysis oil combustion in the SI engine is purposeful. According to a review of the literature, pyrolysis oil requires further in-depth research due to the diversity of its chemical composition resulting from the type of feedstock and parameters of the thermal process.

On the basis of a literary survey, one can highlight that pyrolysis oil can be managed as an engine fuel, thus, investigation in this field is fully justified. Problems with potentially relatively high sulphur content in a raw pyrolysis oil can be solved by applying blends with sulphur-free fuels. Thus, after its deep investigation, a raw pyrolysis oil is potentially a valuable liquid that can be used as a fuel additive or single fuel for the IC engine. In addition, as can be seen from a review of the literature, pyrolysis oil was typically used in amounts no greater than 20% in a classical internal combustion engine. On the other hand, a strategy for the development of internal combustion engines indicates a technology of gasoline compression ignition (GCI) as a promising project. The GCI can lead to a significant increase in engine-indicated efficiency by over 40% and reduce toxic exhaust components soot and NOx, as investigated by Zyada et al. [18]. Moreover, Zhang et al. tested various high-reactivity gasolines and compared them with 91 E10 gasoline [19]. They found similar penetration for these fuels.

As well-known, modern gasoline and diesel internal combustion engines are equipped with high-pressure fuel systems, it is important from the research point of view to test the fuel stream injected by the high-pressure injector and to perform a comparative analysis of the tested liquid with other conventional fuels. The Sauter Mean Diameter (SMD) is used to quantify the injection phenomenon. Many scientific papers discuss the purposefulness and research effects of the SMD-based method. For example, Martinez et al. describe a new optical method for analyzing high-pressure injected fuel sprays in correlation with the SMD parameter [20]. Chen et al. formulated the correlation of the fuel spray SMD with physical parameters, e.g., viscosity, fuel injection pressure, and air-blast pressure [21]. They found the most influential factor in the SMD was fuel injection pressure. Altaher et al. observed that the higher air velocity inside the turbine chamber caused smaller SMDs, which improved the mixing and atomizing of fuel injection [22]. Other interesting works including SMD analysis were realized by Park et al. [23] and Zhou et al. [24]. Valuable research work on spraying and mixture formation with the aid of optical measurement instrumentation was presented by Tzanetakis et al. [25] and Park et al. [26]. They studied the non-reacting spray characteristics of gasoline and diesel fuel injected by high-pressure injectors.

In summary, analysis based on optical techniques and, in particular the SMD parameter, is considered a useful and effective method for analyzing and comparing spray characteristics from engine injectors including the impact of fuels. The tests described in this article are innovative due to the study of the course of fuel atomization by the high-pressure injector, which has a significant impact on the course atomization, vaporization, mixing ignition, and flame development.

As concluded from a study of the literature, there is a lack of knowledge in this field regarding the combustion of mixtures of pyrolysis oil in amounts above 25% with typical hydrocarbon fuels in both the SI GCI (gasoline compression-ignition) engines. Hence, the main goals of this research work are: conducting tests in the field of liquid fuel spraying and atomization, SMD calculations, and statistical analysis of the injection process.

The research described in this manuscript is focused on a comparative analysis of spraying distilled pyrolysis oil (DPO) and gasoline with the main aim of characterizing the spraying process of this pyrolysis oil injected by a high-pressure engine injector. The results can be considered a potential introduction for applying the DPO for the GCI engine.

## 2. Materials and Methods

### 2.1. Technology for Pyrolysis Oil Formation

A unique liquid feed pyrolysis reactor (invention disclosure at Michigan Technological University, Office of Innovation and Commercialization, Houghton, MI, USA) was designed in order to have controllable and rapid heat transfer and tunable vapor residence time within the reactor (Figure 1) [27]. Experimental methods for the pyrolysis system have been described previously by Byrne et al. [28] and Kulas et al. [27]. The liquid feed was achieved by melting and mixing HDPE obtained from Idaho National Laboratory (research collaborator) with a pyrolysis wax solvent in the dissolution tank at a 1:1 ratio [29]. The

pyrolysis conditions for this study were a reaction temperature of 575 °C and a vapor residence time of approximately 1 s under ambient pressure. The pyrolysis vapors were separated into three groups using a dual condenser system. The condenser 1 product is a wax (mostly >C20 linear terminal alkenes), the condenser 2 product is a lighter, liquid product (mainly C5–C20 alkenes), and the gas product is composed of C1–C4 alkenes with a trace of hydrogen. A batch distillation was used to remove heavier components (>C15) from the Condenser 2 liquid product from Tank 2 (250 mL starting material) to improve its suitability as a liquid vehicular fuel. The distilled product was condensed using a water-cooled jacket condenser and collected in 15 mL fractions. Each fraction was then analyzed using GC/MS. Once it was confirmed that each fraction was within the desired range of hydrocarbons (C6–C15), all the fractions were combined into a final liquid product to be used for fuel testing (approximately 150 mL total).

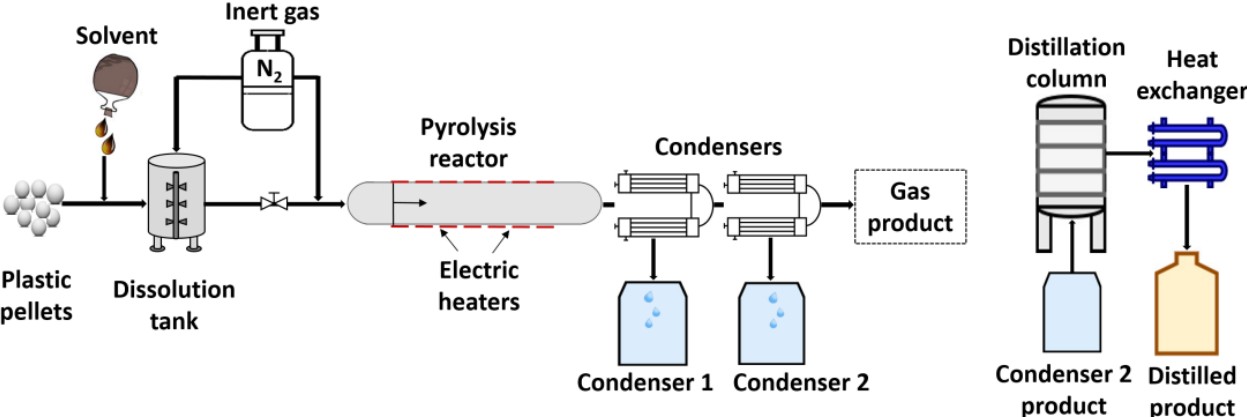

**Figure 1.** Schematic of plastic pyrolysis apparatus and batch distillation column.

## 2.2. Methods for Gas Chromatography–Mass Spectrometry (GC/MS)

The GC/MS methods for analyzing pyrolysis samples have previously been described in Byrne et al. [28] and Kulas et al. [27]. Each pyrolysis sample was run in duplicates.

## 2.3. Spray Analysis

The analysis is based on fuel spraying characterization via SMD. SMD was determined through the application of Malvern Panalytical's Spraytec. This measurement system uses a laser beam and several optical phenomena (e.g., diffraction) that can measure particles and their size. Measurements are based on the laser beam intensity passing through an injected fuel cloud. Next, the analysis of the measurement data is performed, hence, droplet size distribution in real time can be achieved. More detailed information on the system construction and its working principles can be found in the manufacturer's web page [30]. As regards the experimental investigation, each test was repeated 10 times. Log-normal Gaussian distribution of the droplet diameter was determined over time of injection starting from the injection open to its closure. Next, the averaged SMD was determined for each test.

The points which were selected for the analysis are the following:

- In the middle of the injected jet;
- At the edge of the jet.

These points were located at the plane perpendicular to the flow axis at a distance of 70 mm from the nozzle as depicted in Figure 2a. Figure 2b presents an exemplary picture taken by the fast-speed digital video camera installed on the test bench during conducting spraying tests for this work. The test bench is outlined in Figure 3. As shown, the main components are: Spraytec device with the laser beam having a wavelength of 632.8 nm, video acquisition system with a fast-speed video camera (1000 fps, Chronos CR14-1.0, lens: computer 12.5–75 mm f/1.2 zoom lens), and a high-pressure injector with its fueling and control systems. Both fuels (gasoline and distilled pyrolysis oil (DPO)) were injected under

the injection pressure of 400 bar into ambient gas at standard pressure and temperature conditions (1 bar and 20 °C) by the same injector.

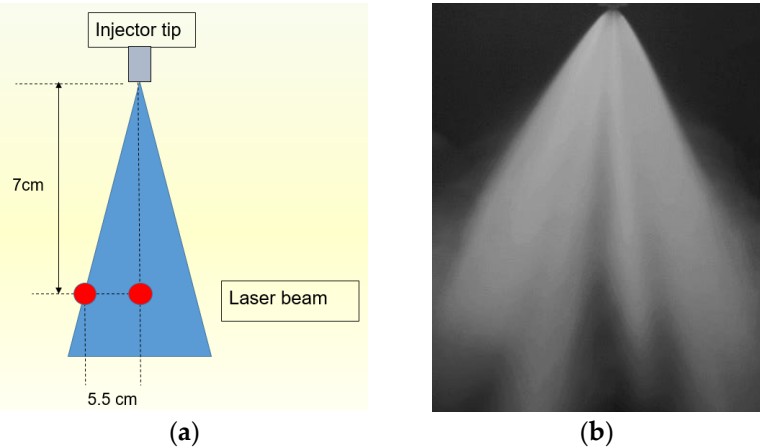

|     |     |
| :-: | :-: |
| (**a**) | (**b**) |

**Figure 2.** (**a**) Location of the points for measuring spray distribution, (**b**) exemplary picture of the injected fuel spraying cloud.

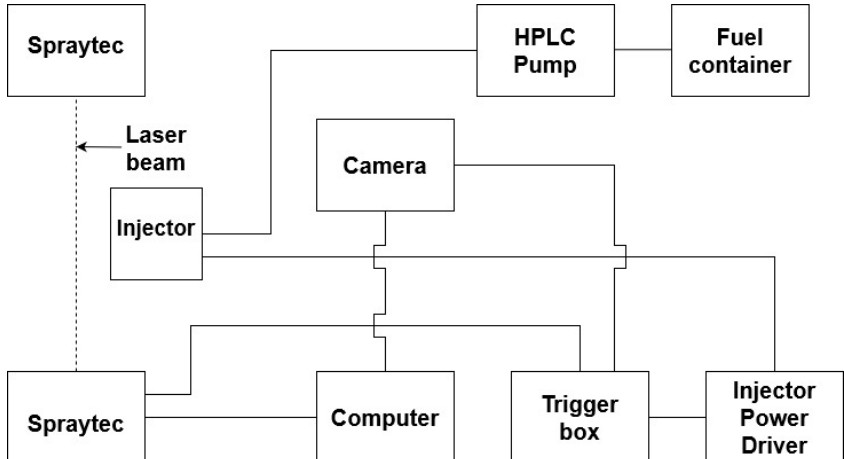

**Figure 3.** Test bench for spray analysis.

### 2.4. Uncertainty and Repeatability of Spraying Tests

Each SMD measurement was a measurement averaged over 10 injections. Each measurement series was repeated 10 times. The number of 100 injections was a sufficient population to calculate the standard deviation (STD) according to the normal distribution. The STD for the average value did not exceed 0.9 μm and they were presented in the charts as boundary lines for the trend channel. The charts show selected SMD courses for individual single injections to visually show their unrepeatability.

## 3. Results

### 3.1. Results from Chemical Analysis

The pyrolysis oil obtained from the pyrolysis of HDPE was distilled with respect to removing heavy hydrocarbons; C16–C27. Chromatographs presenting its chemical compositions are depicted in Figure 4A for raw pyrolysis oil and Figure 4B for the distilled oil. As observed, there is a significant difference in its chemical composition. The DPO contains light and medium hydrocarbons C6 to C15. The heavier hydrocarbons (over C15) were removed, thus, their concentration in the distilled oil can be considered marginal.

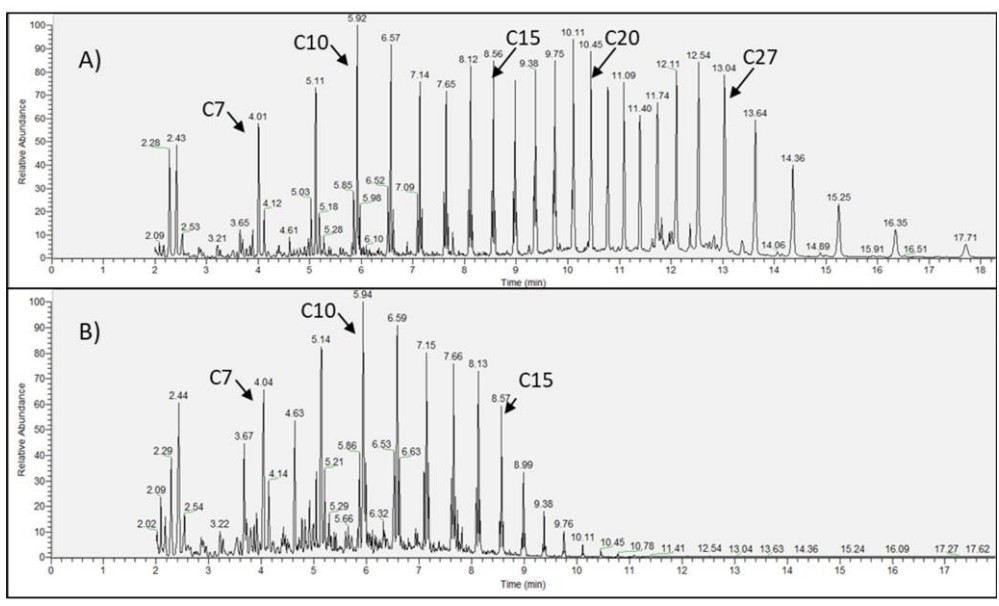

**Figure 4.** Chromatographs for (**A**) raw pyrolysis oil and (**B**) distilled pyrolysis oil.

A closer look into the percentage of the hydrocarbons C6–C15 in the distilled oil is shown in Figure 5. As found, the majority are the compounds C6–C11, which are present in amounts of nearly 70% in this oil. Figure 5b presents the typical gasoline HC content by Chevron [31]. Comparing both graphs, it can be seen that DPO contains much more C11–C15. This is numerically about 30%. Hence, it can be concluded that the share of C11–C15 will significantly affect the easier self-ignition DPO, Therefore, one can conclude this HC content can provide premises for applying this oil as a potential substitute fuel for gasoline in either low-compression SI or GCI engines.

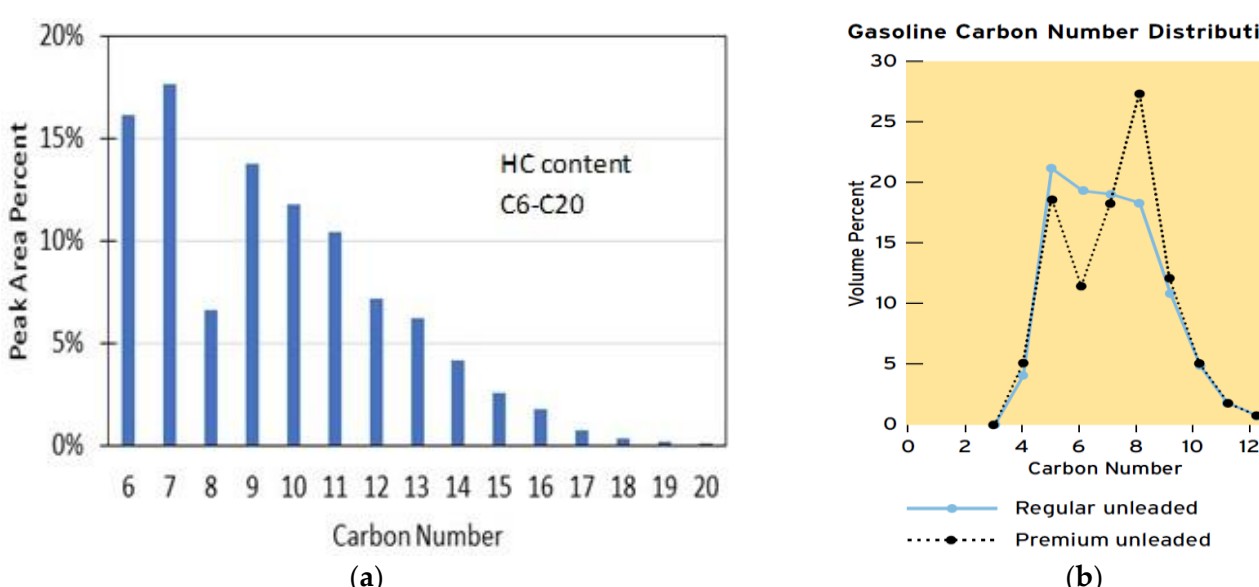

**Figure 5.** HC content in (**a**) DPO and (**b**) gasoline [31].

### 3.2. Results from Spraying Tests

As mentioned, the analysis was realized as the comparative analysis presenting differences in SMD for the DPO and 91-RON gasoline. Figure 6a,b presents the SMD for the distilled pyrolysis oil, and Figure 6c,d for gasoline, respectively. There are two histograms for both of these fuels. Histograms in Figure 6a,c present particle distributions expressed by Volume Frequency (pdf) at the beginning of injection, whereas histograms in Figure 6b,d

show particle distribution at the end of injection. Additionally, the cumulative density function (cdf) is represented by the cumulative volume. In order to investigate possible changes in the size of the atomized fuel droplets, the SMD parameter was calculated for the time period between 2 and 5 ms from the time of opening the injector and starting the injection. The particle size distributions are in the range of 2.8 to 28 µm. It can be observed that there are no significant changes in the distribution of the fuel droplet diameters for the two fuels. Slight differences are noticeable for the mean droplet diameters during injection development between the fuels. These changes, however, can be considered marginal and do not significantly affect the evaporation, premixing, and forming of the combustible mixture. The observed trend seems to be horizontal and is confirmed by the results presented in Figure 7a,c.

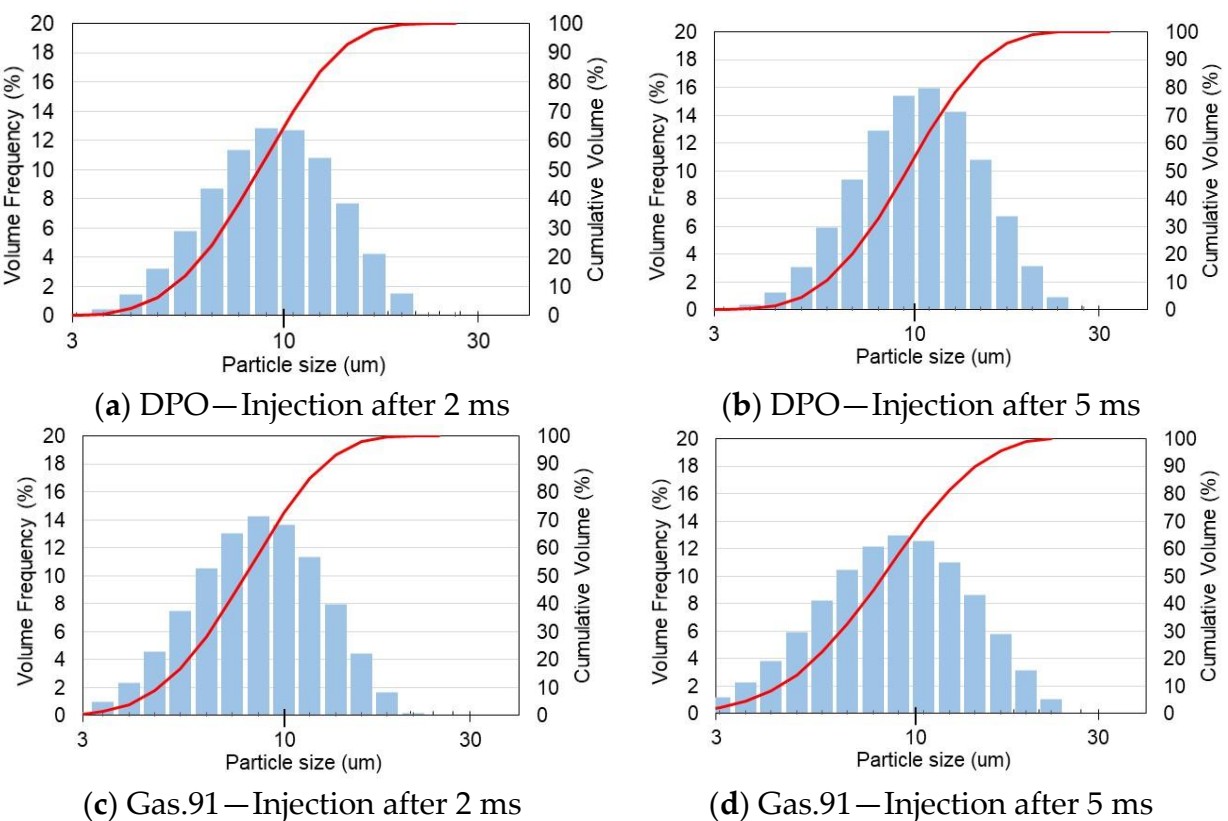

(**a**) DPO—Injection after 2 ms

(**b**) DPO—Injection after 5 ms

(**c**) Gas.91—Injection after 2 ms

(**d**) Gas.91—Injection after 5 ms

**Figure 6.** Averaged particle size distribution for the edge points at the beginning (**a**,**c**) and the end (**b**,**d**) of spraying for pyrolysis oil (**a**,**b**) and gasoline (**c**,**d**).

Interestingly, the average pyrolysis oil droplet SMD is slightly larger than the 91-RON gasoline droplet size by nearly 1 µm, as observed in Figure 7, for both points located centrally and at the edge of the injected fuel stream. Furthermore, an increase in the SMD parameter was observed (Figure 7b,d) with the evolution of injection. This can be explained by droplets merging into larger droplets as a result of collision and coalescence. It is very probable, due to the high concentration of the atomized fuel droplets in the center of the stream and the high flow velocities of the droplets at the injector nozzle catching up droplets injected earlier. The SMD determined for central points is also higher for DPO in comparison to the 91-RON gasoline. The dashed lines show the trend channels in both analyzed cases of the location of the measurement points.

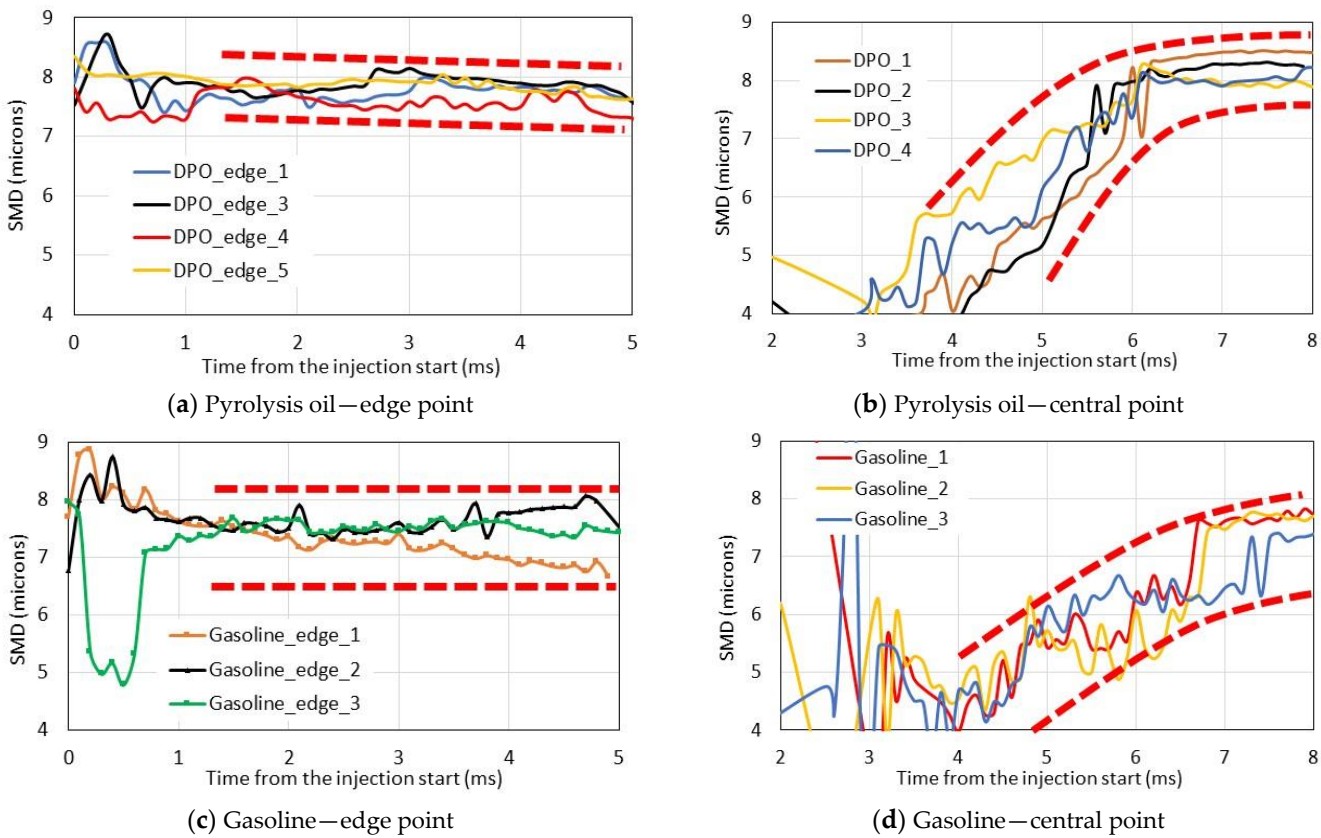

**Figure 7.** Time histories for averaged SMD for pyrolysis oil (**a**,**b**) and gasoline (**c**,**d**) measured at the edge and the central point.

## 4. Discussion

The main research task was to assess the quality of DPO injection, spraying, and atomization by the high-pressure injector used to directly inject gasoline into a spark-ignition engine. This analysis was performed as a comparative analysis also performed for 91-RON gasoline, which is commonly used in the US. The SMD was adopted as a parameter for the comparative analysis. The SMD parameter reliably assesses the fineness of the atomized fuel droplets. It was recognized that the tests can be carried out in an environment that differs from the actual conditions of pressure and temperature inside the engine cylinder during its operation. Thus, the tests were carried out at an atmospheric pressure of 1 bar and at a room temperature of 21 °C. The actual pressure in the SI engine cylinder is in the range of 15–25 bar and the temperature can be in the range of 200–300 °C depending on both the compression ratio and boosting this engine. Therefore, a comparative analysis was adopted as the research method. It was considered that the difference between the injection pressure (400 bar) and the environmental pressure (15–25 or 1 bar) will not significantly affect the quality of the atomized fuel; unlike temperature, which significantly affects the evaporation rate of atomized fuel droplets. In this case, the comparative analysis shows differences in droplet size for both fuels, and it can be assumed that these differences will not change significantly at elevated temperatures.

Regarding SMD analysis, the averaged SMD for DPO is slightly higher (by 1 μm) in comparison to the 91-RON gasoline. This difference can be considered crucial in the evaporation process of the fuel droplets. Bigger droplets need a longer time for their complete evaporation. Assuming the droplet diameter is higher by 12%, it is associated with its surface increase by 25%, hence, according to the D2 law [32], the time for evaporation will also increase in nearly the same ratio. However, if the injection is realized as a port-fueled strategy, then the evaporation process can be considered insignificant. On the other hand, direct-injection evaporation and premixing phenomena inside the engine cylinder

can be controlled by injection timing, increased injection pressure, and/or change in the injector nozzle configuration. Summing up, one can state that the differences in atomization of both liquids are not so significant that it would require a change in the injection pressure or the use of a different injector design or injection technology.

## 5. Conclusions

The following conclusions can be drawn from this analysis:

- Waste plastics consisting of HDPE can be applied as a feedstock for the pyrolysis process with its main objective being to produce pyrolysis oil.
- The distillation process successfully removed heavier hydrocarbons and makes the pyrolysis oil in carbon number distribution for gasoline-fueled engines, either spark-ignited or GCI engines. Finally, the hydrocarbons C15–C30 were significantly reduced after the distillation.
- The Sauter Mean Diameter can be considered the parameter that can reliably characterize the spraying and atomization of distilled pyrolysis oil.
- The SMDs for 91-RON gasoline and DPO were found to be similar to each other, hence, a direct gasoline injection strategy can be implemented. However, as tested, the SMD for DPO was found to be on average 1 μm larger than that of the 91-RON gasoline.
- Due to the high content of C11–C15 compounds that can promote easier self-ignition, DPO from HDPE pyrolysis can be considered a single fuel or a potential additive for blending the gasoline 91 to form a highly reactive fuel for the compression-ignition strategy in the reciprocating engine.
- Regarding the injection process, there are no remarkable drawbacks to applying DPO as a substitute fuel for direct injection into the internal combustion engine.

**Author Contributions:** Conceptualization, M.S., J.D.N., D.S. and S.S.; methodology, M.S., J.D.N. and D.S.; software, M.S. and S.S.; validation, A.Z., D.K. and D.S.; formal analysis, J.D.N. and D.S.; investigation, M.S., A.Z. and D.K.; data curation, M.S., A.Z. and D.K.; writing—original draft preparation, M.S.; writing—review and editing, M.S., J.D.N., D.S. and S.S. All authors have read and agreed to the published version of the manuscript.

**Funding:** This work was partially supported by the Polish National Agency for Academic Exchange Iwanowska program PPN/IWA/2019/1/00149/U/00001, investigation of liquid products from thermal processing of waste as fuel to the internal combustion engine.

**Data Availability Statement:** Not applicable.

**Acknowledgments:** Special thanks for the help provided by the technical and support staff of the APSRC labs in the ME-EM and Chemical Engineering departments of Michigan Technological University, especially Henry Schmidt and William Atkinson.

**Conflicts of Interest:** The authors declare no conflict of interest.

## Abbreviations

| | |
|---|---|
| CI | compression ignition |
| CO | carbon monoxide |
| CR | compression ratio |
| DCM | dichloromethane |
| DPO | distilled pyrolysis oil |
| GCI | gasoline compression ignition |
| GC/MS | Gas Chromatography—Mass Spectrometry |
| HC | hydrocarbons |
| HDPE | high-density polyethylene |
| NOx | nitric oxides |
| RON | research octane number |

SI      spark ignition
SMD     Sauter mean diameter
STD     standard deviation

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
