# Peer review of "Comparative Analysis of Injection of Pyrolysis Oil from Plastics and Gasoline into the Engine Cylinder and Atomization by a Direct High-Pressure Injector"

_energies, doi:10.3390/en16010420_

Round 1
Reviewer 1 Report
Thank you so much for submitting your work to the journal of Energies. The overall very high-quality research paper shows strong expertise in this field. This manuscript could be published after considering some minor changes as below:
1. Title: It would be good to revise the title, high-pressure injector for what application? The title has to attract researchers to read the paper related to any application. Is this research lead to lower emissions? Is anything related to Low emission vehicles? Zent-zero agenda? Please try to sell your writing.
2. Abstract: Please avoid using the abbreviation for the first time without explaining it. for example HDPE
3. Keyword: Please revise them and avoid repetition of pyrolysis
4. Introduction: Please avoid having reference lumps [1-4]; references have to be cited at the end of the sentence one by one.
5. Can you please explain in 1-2 sentences how repeatable are your results? how did you control uncertainty and variation of data?
Author Response
The file is attached.

Reviewer 2 Report
The topic of the article is quite interesting. However, in my opinion, it is not suitable for publication in its current form.
Below are my comments:
1. Too long an introduction compared to the description of the fuel production method and the fuel atomization test. The introduction should be shorter, but the "Materials and Methods" section should be expanded.
2. I believe that such a lengthy description of tire fuel for diesel engines with the title of the article is not needed. I don't understand what this has to do with the article's title.
3. In the "methodology" section, the fuel spray test method description is missing. It should be supplemented. The "Materials and Methods" section should also be supplemented with a description of the measuring station and the type of test injector used. In its current form, it is not
4. It is not clearly stated whether the DPO and 91-RON tests were carried out on the same injector. This should be marked and described
5. Lack of scientific and factual discussion. Add a Discussion section. In my opinion, the description of the results and the discussion in one chapter are unacceptable "Results and Discussion". Moreover, the description of the "Results" and the "Discussion" was described in only five lines of text.
6. In addition, the conclusions section should be developed. In the current version, there are too few of them, and they are not sufficient.
I recommend improving the article.
Author Response
The file is attached.

Reviewer 3 Report
The article "Comparative analysis of plastic pyrolysis oil spray by the high-pressure injector" presents a study relevant to the development of the use of alternative fuels in internal combustion engines. The essential tested parameter was Sauter Mean Diameter (SMD) of fuel droplets measured at the high injection pressure.
Notes:
- The article should include a description of the abbreviations.
- More chemical and physical properties of plastic pyrolysis oil should be provided.
Author Response
The file is attached.

Round 2
Reviewer 2 Report
Dear authors,
thank you for considering my comments.
I believe that the article in its present form is suitable for publication.
Reagrds